# Evaluation of the Presence of Arbuscular Mycorrhizae and Cadmium Content in the Plants and Soils of Cocoa Plantations in San Martin, Peru

Bernabé Luis-Alaya [1], Marcia Toro [1,2,*], Rocío Calsina [1], Katty Ogata-Gutiérrez [1], Alejandra Gil-Polo [1], Ernesto Ormeño-Orrillo [1] and Doris Zúñiga-Dávila [1]

[1] Laboratorio de Ecología Microbiana y Biotecnología, Departamento de Biología, Facultad de Ciencias, Universidad Nacional Agraria La Molina, Lima 15026, Peru

[2] Centro de Ecología Aplicada, Instituto de Zoología y Ecología Tropical, Facultad de Ciencias, Universidad Central de Venezuela, Caracas 1041-A, Venezuela

[*] Correspondence: mtoro@lamolina.edu.pe or marcia.toro@ucv.ve; Tel.: +58-414-3056459

**Abstract:** Cocoa (*Theobroma cacao* L.) is an important crop in Peru. International regulations require products derived from cocoa to be free of heavy metals (HMs), such as cadmium. Arbuscular mycorrhizae (AM) contribute to reduced HM content in the plant, preventing its accumulation in the fruit and facilitating the rhizodeposition of HMs through glomalin-related soil proteins (GRSP). We studied the presence of mycorrhizal symbiosis in cocoa plants and cadmium in three plantations in San Martín, Peru. The maximum Cd content detected in soils was 1.09 (mg/kg), an amount below the tolerable limit for agricultural soil ($\geq$1.4 mg/kg). Cocoa roots showed 68–86% active mycorrhizal colonization; agronomic management did not cause differences between plantations. Levels of GRSP were between 7.67 (GRSP-EE) and 13.75 (GRSP-T) mg protein g soil$^{-1}$. Morphological and molecular analysis of Glomeromycota fungi showed the presence of families Claroideoglomeraceae, Paraglomeraceae, Gigasporaceae, Glomeraceae, Acaulosporaceae, Archaeosporaceae, and Diversisporaceae. Our results show the presence of arbuscular mycorrhizal symbiosis in cocoa plantations and suggest that *T. cacao* may phytostabilize HM in its rhizosphere through the production of GRSP. The presence of mycorrhizal symbiosis indicates the potential for the preparation of biofertilizers for cocoa since the production of GRSP is promissory for the biostabilization of soil HMs.

**Keywords:** cadmium; cocoa; glomalin-related soil proteins; *Theobroma cacao*





## 1. Introduction

Peru is one of the leading exporters of organic cocoa beans in the world [1]. The Peruvian production of cocoa seeds has been increasing steadily for ten years, growing at a rate of 12.6% on average per year. More than 90,000 families depend economically on this activity, which has positioned Peruvian cocoa favorably in the international market [2]. The bioaccumulation of heavy metals by cocoa plants affects the quality of the exported product. The uptake of cadmium (Cd) has recently raised public concern since high concentrations of this metal have been found in seeds from cocoa trees grown on soils with intermediate Cd levels [3] and in final cocoa products [4]. Cd concentrations in the soils of the Peruvian Amazon show up to 3.54 mg/kg, with an average of 1.56 mg/kg. In addition, 39% of the territory is above the tolerable limit for Cd in agricultural soil ($\geq$1.4 mg/kg), as indicated by Rojas et al. (2021) [5].

Since market-based studies found chocolate products containing Cd concentrations with mean values above 100 $\mu$g kg$^{-1}$ (equivalent to 0.1 mg kg$^{-1}$), the European Union (EU) established regulations for critical values of Cd in cocoa products imported to the EU, which came into force in 2019 [6,7]. Cocoa products produced with beans from Latin America generally contain high Cd concentrations, with variations among different provenances.

In Ecuador, an average of 0.94 mg kg$^{-1}$ has been detected in cocoa seeds [8]. In Peru, amounts over 0.5 mg kg$^{-1}$ have been found in cocoa seeds [9] and 1.51 mg kg$^{-1}$ has been found in different parts of cocoa plants in Amazonian plantations [10]. The data illustrate the problems faced by Peruvian cocoa for consumption and export in the international market [9,11]. Cd in soil has been primarily attributed to different soil parent materials, input from phosphorus (P) fertilizers, and/or soil properties affecting the availability of soil Cd to plants [4]. Ameliorating the Cd content in cocoa must be tackled to maintain the quality of Peruvian cocoa and maintain its acceptance in the international market.

Due to the need to reduce the Cd content in cocoa, interest has arisen in using beneficial microorganisms, such as arbuscular mycorrhizae fungi (AMF). AMF positively affect plant nutrition, development, and protection against biotic and abiotic stresses. The presence of heavy metals (HMs) in the soil, such as Cd, Pb, Zn and Cu, among others, constitutes an abiotic stress to which mycorrhizae help the plant mitigate [12,13]. AMF accumulate HMs in the plant roots, thus minimizing their content in the fruit [14], or immobilize HMs in the rhizosphere through the production of proteins as glomalin [15]. This process, known as phytostabilization, decreases the bioavailability of these elements and therefore, their toxicity [16,17]. Glomalin also stimulates agglutination of particles to form aggregates that improve soil structure [18,19]. Moreover, AMF can also affect the transformation of trace metals in the rhizosphere through acidification, modification of root exudates, or hyphal sequestration [20].

Cocoa has long been known to be a highly mycotrophic plant [21]. Several authors have reported AMF colonization in cocoa plantations under different agricultural management methods [22–24], suggesting that symbiosis exerts an important role in cocoa nutrition and agrosystem nutrient cycling. This crop is usually grown in greenhouses before being planted in the field [25,26]. This management facilitates the inoculation of plantlets with AMF, allowing the plant to establish in the field with a reinforced root that may help the plant withstand abiotic and biotic stresses, thereby obtaining better fruit quality. Iglesias et al. (2011) [27] found that cocoa plants had good growth and development thanks to the high compatibility and performance of native Glomeromycota fungi inoculation. On the other hand, in addition to being better adapted to the edaphoclimatic conditions of the area, native AMF species ensure the success of functioning symbiosis, improve the development of plants, and potentiate the conservation of the local AMF community [28].

AMF isolated from contaminated areas cope better with metal toxicity than those isolated from unpolluted soils, and they are thus potential candidates to be used in strategies to mitigate Cd soil contamination [29]. Ferrol et al. (2009) [30] suggested that indigenous, and presumably adapted AMF, were more suitable for phytoremediation purposes. They could serve as a potential biotechnological tool for successfully restoring degraded ecosystems. Therefore, it may be appropriate to apply Glomeromycota fungi inocula to cocoa in the nursery before being taken to the field.

In order to explore the presence of AMF in Peruvian cocoa fields and with the aim of diagnosing the prevalence of symbiosis associated with cocoa plants, three organic fields located in Tarapoto, San Martin, Peru, were sampled to assess mycorrhizal colonization, identify Glomeromycota fungi, and study soil proteins related to glomalin and its bond with Cd abundance in soil. We expected to find the presence of arbuscular mycorrhizae associated with cocoa plants in the organic plantations of San Martín. Our main interest is to generate knowledge about native AMF and its potential for the manufacture of biofertilizers to inoculate cocoa plants in Peruvian fields.

## 2. Materials and Methods

### 2.1. Description of the Area of Study

Three organic cocoa plantations located in the department of San Martín, Province of Tarapoto (76°31′ 33.69′′ W, 6°23′ 55.33′′ S, and 791 m.a.s.l.) were studied. Close to 90% of cocoa production in Peru is found in the Amazon area, where the San Martín region ranks first with 42% of production, constituting an inclusive activity as it is carried out

mainly in former coca-growing areas, native communities, and borders [31]. The year the sampling was carried out, the region had an average temperature of 25.5 °C, 70% relative humidity, and average monthly precipitation of 69 mm. The plantations were 3, 9, and 10 years old; the seedlings in the nursery grew in an organic substrate (compost). In the oldest plantation, foliar fertilization was eventually applied. The plantations had diverse varieties of cocoa with the predominance of clone CCN-51, which was selected to conduct this study. Clone CCN-51 is recognized for its high yield, adaptability to different regions and environments, and remarkable productivity, with estimated yields of 1300–1800 kg/ha. This clone originated in Ecuador and gradually expanded through the cocoa-production regions of Ecuador, Colombia, Brazil, and Peru [32].

### 2.2. Sampling Description: Rhizosphere and Plant Analysis

We carried out sampling in the fields described above at plantations that were 3, 9, and 10 years old. Given the predominance of cocoa plants as the only crop in the plots, we carried out random sampling. In each plot, three 1 kg samples of the rhizospheric soil of a cocoa plant (0–20 cm) were randomly taken with a small shovel, consisting of roots and the surrounding soil, according to the definition of rhizosphere of McNear Jr. (2013) [33]. These samples were stored at 4 °C for immediate biological analysis. Live roots of cocoa plants were taken from each of the samples to perform vital staining and observation of live mycorrhizae. A portion of each soil sample (300 g) was taken to count Glomeromycota spores and determine soil proteins similar to glomalin, as described below. The remaining soil sample was used to establish trap pots in order to reproduce the native spores of the Glomeromycota fungi from each plantation and study their diversity. From the same plants chosen for rhizosphere sampling, leaves and seeds of cocoa pods were taken to analyze the HM content at the Laboratory of Soils of UNALM. For statistical analyses, all parameters mentioned so far were analyzed in triplicate.

### 2.3. Soil Analysis

For the chemical analysis of the soil, 5 samples of bulk soil (0–20 cm) were taken in each field and mixed to obtain a composite sample, which was dried and sieved (2 mm). The Laboratory of Soils at UNALM made the following chemical analyses of soils: pH and electric conductivity (1:1), organic matter (% O.M.) according to Walkley and Black (1934) [34], available P (Olsen) and available K (ammonium acetate) as stated by Jackson (1995) [35], Al [36], and HM in soil (Pb, Cr and Cd) and plant (Cd) by method 3052 EPA [37] (Table 1).

**Table 1.** Chemical analysis of soil, leaves, and seeds from the three cocoa fields studied in San Martin, Tarapoto.

| Cocoa Field | pH | Salinity (dS/m) | Organic Matter (%) | P | K | Al | Pb | Cd | Cr | Cd Leaves | Cd Seeds |
|---|---|---|---|---|---|---|---|---|---|---|---|
| | | | | | | | | (mg/kg) | | | |
| 3 years | 3.88 | 0.18 | 2.32 ** | 2.3 * | 54 * | 1.2 | 17.64 | 0.86 | 12.24 | - | - |
| 9 years | 4.21 | 0.12 | 3.56 ** | 13.5 *** | 72 * | 0.65 | 14.18 | 1.09 | 12.96 | 1.58 | 0.3 |
| 10 years | 4.71 | 0.1 | 2.36 ** | 22.7 *** | 71 * | 0.2 | 15.69 | 0.37 | 10.04 | 2.00 | 0.63 |

* Low ** Medium *** High (according to the interpretation given by the soil laboratory of the UNALM).

### 2.4. Installation of Trap Pots

To multiply the spores of Glomeromycota fungi, two trap pots (1 kg each) were established with rhizospheric soil samples from each cocoa plantation, with *Brachiaria decumbens* (50 seeds/pot) pre-germinated for a total of 6 trap pots. They were established for 5 months, after which the number of spores was evaluated, and their propagation continued in the same way for 2 more cycles of 5 months each. During propagation, the temperature and humidity oscillated from 13.3 to 35.6 °C and 33% to 90%, respectively. For evaluation, 50 g of rhizospheric soil was taken, the wet sieving and decanting method was applied [38] with subsequent centrifugation, and a sucrose gradient [39] was applied for the separation and quantification of spores and subsequent mounting and microscopic observation.

### 2.5. Vital Root Staining and Quantification of AMF Colonization

Root samples were taken in triplicate from each cocoa field, immediately transferred to the laboratory, and stored at 4 °C for the application of vital staining and observation of the living structures of the AMF, according to Schaffer and Peterson (1993) [40]. The samples were incubated for 24 h in the dark at 37 °C in a solution containing 2.5 mL Tris buffer (pH 7.4) and 2.5 mL Nitro Blue Tetrazolium (1 g in 250 mL deionized water).

The roots were washed with abundant distilled water and placed in 10 mL of saline formalin at room temperature in the dark, rinsed 2 to 3 times with deionized water to remove the saline formalin, transferred to glass test tubes, and boiled in 10 mL of chloral hydrate for 1.5 h After this, the roots were rinsed two to three times with deionized water and incubated in 10 mL of ammonium peroxide solution for 4.5 h in the dark with shaking. Subsequently, they were incubated in 10 mL of KOH at 55 °C for 24 h. The roots were rinsed 2–3 times in deionized water, immersed in 1% HCl (*v/v*) for 5 min, and incubated without rinsing in acid fuchsin solution in the dark at 55 °C for 2 h.

For quantification, stained root segments were mounted on slides with a drawn grid in the reverse. At each intersection of the roots, hyphae, coils, arbuscules and/or vesicles present in the root with a line of the grid were quantified using the methodology described by Giovanetti and Mosse (1980) [41]. The percentage of colonized root length was calculated as follows:

Active colonization = (all active structures)/(number of intersections counted)
Total colonization = (all mycorrhized structures)/(number of intersections counted)

This methodology enabled quantifying the amount of active and functional structures of the AMF.

### 2.6. Determination of Glomalin-Related Soil Proteins and Bradford Method

We applied the method described by Wright and Upadhyaya (1998) [42], in which a soil sample is subjected to successive extractions with sodium citrate buffers at two concentrations to obtain the proteins contained in the soil, subjecting the sample to autoclave cycles. Thus, the extracts containing fractions of GRSP-EE and GRSP-T were obtained and stored at 4 °C to determine the protein content. Subsequently, aliquots of each of the sodium citrate extracts for GRSP-EE and GRSP-T were taken, and the methodology of Bradford (1976) [43] was applied. This method consists of determining the amount of proteins present in the sample by reaction with Coomassie blue dye, using a calibration curve constructed with bovine serum albumin (1 mg/mL). The results are expressed as mg protein/g of soil, according to Yang et al. (2017) [44]. The equation of the calibration curve was $y = 0.0458x - 0.0037$, $R^2 = 0.985$. The analyses were carried out in triplicate in each field studied.

### 2.7. DNA Extraction of Glomeromycota Fungi

For the molecular characterization of the AMF, samples of mycorrhizal propagules from the trap pots with *Brachiaria decumbens* and the rhizospheric soil of the cocoa plants were used. AMF spores were collected from approximately 1 g of soil sample, as indicated by Faggioli et al. (2019) [45]. Under a stereoscope, we cleaned the samples to get rid of debris and plant material. The cleaned spores were transferred to a 1.5 mL Eppendorf tube and water was removed with a pipette tip. Then, the spores were frozen with liquid nitrogen and crushed using a sterile micro-pestle in solution C1 from the DNAse® PowerSoil® kit (Qiagen Inc., New York, NY, USA). Then, the manufacturer's instructions were followed to purify the DNA.

### 2.8. PCR, Cloning and Sanger Sequencing of Glomeromycota Fungi

The 18S rRNA gene was amplified using the primers AML1 and AML2 [46]. PCR was performed using Phusion High-Fidelity PCR Master Mix plus DMSO (Thermo Fisher Scientific, Inc.) using 10 ng of DNA (up to 5 µL) per sample. The reactions were run as follows: 3 min of initial denaturation at 96 °C, 30 cycles of 96 °C for 1 min, 58 °C for

30 s, and 72 °C for 1 min, followed by 72 °C for 5 min. To insert 3′ A overhangs, we added 0.2 μL of TaqDNA polymerase, recombinant (5 U/μL) (Thermo Fisher Scientific, Inc., Waltham, MA, USA) to the PCR tubes immediately after PCR was finished, and incubated the tubes in a thermal cycler set up at 72 °C for 10 min. The products were examined on 0.7% agarose gel and cleaned using the GeneJET gel extraction kit (Thermo Fisher Scientific, Inc.). Cloning was performed using the TOPO®TA Cloning® Kit for sequencing (Thermo Fisher Scientific, Inc.) with a vector: insert molar ratio of 1:4. The mixture was chemically transformed into *E. coli* TOP10 and transformants were then selected by incubation in Luria-Bertani (LB) medium plates containing kanamycin (50 μg/mL) and X gal (80 μg/mL) at 37 °C overnight. White colonies were selected and colony PCR, with primers M13Fwd and M13Rev, was performed to confirm that the transformants were carrying the insert. The plasmids were extracted with GeneJET Plasmid Miniprep Kit (Thermo Fisher Scientific, Inc.). The latter was sequenced and bioinformatic analysis was carried out using the NCBI database and corroborated with a special database for mycorrhizal fungi, the MaarjAM online database [47].

### 2.9. Phylogenetics of Glomeromycota Fungi

160 sequences from SSU 18S sRNA belonging to the class Glomeromycetes were used to infer the phylogenetic relationships. *Mortierella polycephala* and *Endogone pisiformis* were set as out-groups. Nucleotide sequences were aligned using the MUSCLE tool in MEGA version 6.0 software [48]. The phylogenetic tree was generated by Bayesian analysis of the dataset using MrBayes version 3.1.2 software [49]. Posterior probabilities (pp) were adjusted for 5,000,000 generations employing the Monte Carlo Markov Chain procedure and four simultaneous tree-building chains (nchains: 4), with every 100th tree saved (samplefreq: 100). The phylogenetic tree was rooted and visualized using FigTree software (http://tree.bio.ed.ac.uk/software/figtree/ accessed on 1 May 2020).

### 2.10. Analysis of Spore Morphotypes

The selected spores isolated from the trap pots were separated by morphotype, counted on a Petri dish, and grouped by size, color, and shape. Next, we placed similar groups on microscope slides with polyvinyl alcohol in lacto-glycerol (PVLG). A second group of spores was assembled with PVLG + Melzer's reagent 1:1 (*v/v*), covered with a coverslip, and gently broken to expose the inner walls mounted on slides with PVLG. The results of the color reaction with Melzer's reagent were used for microscopic evaluation of the structural phenotypic characteristics and taxonomic identification. Species description was conducted according to the identification manual from Schenck and Perez (1990) [50] and online reference species descriptions in INVAM (International Culture Collection of (Vesicular) Arbuscular Mycorrhizal Fungi, http://invam.caf.wvu.edu, 2020 accessed on 1 March 2019). We interpreted the taxonomic characteristics with observations under an optical microscope with (40×) immersion objective (100×).

### 2.11. Statistical Analysis

The results for the parameters AMF colonization and glomalin-related soil proteins was presented by comparing the means and their standard deviation. They were also analyzed using one way ANOVA in SPSS software To verify differences between means, the Duncan test $p < 0.05$ was applied as a posteriori test. Multivariate analysis using Statgraphics Centurion XVI (StatPoint Technologies Inc., Warrenton, CA, USA) software was conducted among all biological variables to obtain Pearson correlation coefficients. *p*-values below 0.05 indicated correlations significantly different from zero.

## 3. Results
### 3.1. Soil Chemical Analysis

The plantation soils (Table 1) had pH values between 3.88 and 4.71, classified as strongly acidic soils (<5.5) by the UNALM Soil laboratory. There was no risk of the

soils being affected by salinity as they were very slightly saline (<2 dS/m), and they had medium organic matter content, although this value was slightly higher for the soil of the 9-year-old plantation than for that of the other two plantations. According to the Ministry of Agriculture of Peru [51], the pH values of the soils in this study were considered tolerable, coinciding with pH values of soils of other localities in the department of San Martín and with the average organic matter content suitable for the cultivation of cocoa.

According to DS N° 011-2017 MINAM [52], the Pb content in agricultural soil should not exceed 70 mg/kg dry soil; the amounts recorded in the Tarapoto plantations ranged between 14.18 and 17.64 mg Pb/kg dry soil. Regarding Cr, the quantities registered (10.04–12.96 mg Cr/kg dry soil) were not considered risky for agricultural soils. The soils in this work did not have a risk of contamination by Pb and Cr and were suitable for agriculture.

### 3.2. Cd in Soil, Cocoa Leaves, and Seeds

The three cocoa plantations showed soil Cd concentrations below the levels allowed by DS N° 011-2017-MINAM in Peru (1.4 mg Cd/kg dry soil) [52]. Therefore, the cocoa products and their derivatives would not be at risk of toxicity in this case. Even though the soils sampled in this work did not have high Cd content, cocoa has a bioaccumulative nature and is able to absorb Cd from the environment by storing it in its tissues without eliminating it through metabolic processes, as shown in Table 1.

### 3.3. Quantification of Glomeromycota Spores and Arbuscular Mycorrhizal Fungi Root Colonization

Table 2 shows that the highest number of spores was observed in the soil of the 3-year-old plantation. The soil of the 9- and 10-year-old plantations had significantly lower amounts of spores than that of the three-year-old plantation. The amounts recorded in this study ranged from 13.4 to 131.8 spores 100 g dry soil$^{-1}$, which were slightly lower than those recorded in studies of cocoa plantations in tropical zones. The colonization of the cocoa roots (Table 2) allowed us to observe the typical structures of the arbuscular mycorrhiza, coils, vesicles, arbuscules, and mycelium. Glomeromycota fungi are characterized by intra- and intercellular growth in the root cortex, forming the structures mentioned above and the mycelium external to the root. The arbuscules are hyphae with dichotomous division, invaginated by the plasmatic membrane of the cortical cells, and have short life spans (Figure 1b). They are essential for the exchange of nutrients and carbohydrates between the fungus and plant [53]. It is common to observe arbuscules of different ages in the same sample. Vesicles are storage structures formed in the terminal part of the hyphae and may last longer in the root than in the arbuscules (Figure 1a). The mycelium external to the root extends to capture nutrients from the soil (Figure 1c). Total mycorrhizal colonization of cocoa in the three plantations had statistically similar values (68–86%) as well as active mycorrhiza between 59 and 63%, indicating the presence of mycorrhizal symbiosis and its functionality in the fields studied.

**Table 2.** Numbers of AMF spores, arbuscular mycorrhizal structures, active and total mycorrhizal colonization, and GRSP-EE content in the roots of cocoa plants in San Martin, Peru.

| Cocoa Field | Number of Spores/100 g Dry Soil | Number of Inactive Hyphae | Number of Active Hyphae | Number of Coils | Number of Active Arbuscules | Number of Vesicles | Total Active Mycorrhizal Colonization (%) | Total Mycorrhizal Colonization (%) | GRSP-EE Content (mg/g Dry Soil) | GRSP-T Content (mg/g Dry Soil) |
|---|---|---|---|---|---|---|---|---|---|---|
| 3 years | 131.8 a | 30 a | 13 a | 9 * | 19 a | 0 | 59 a | 77 a | 8.30 a | 9.58 a |
| 9 years | 65.6 b | 33 a | 18 a | 13 * | 33 a | 9 * | 61 a | 86 a | 8.70 a | 13.75 b |
| 10 years | 13.4 b | 21 a | 19 a | 4 * | 28 a | 0 | 63 a | 68 a | 7.67 a | 12.83 b |

The data represent the mean of three independent replicates with three repetitions each. Means in each column followed by the same letters are not significantly different between fields ($p < 0.05$, Duncan). * Does not have three repetitions.

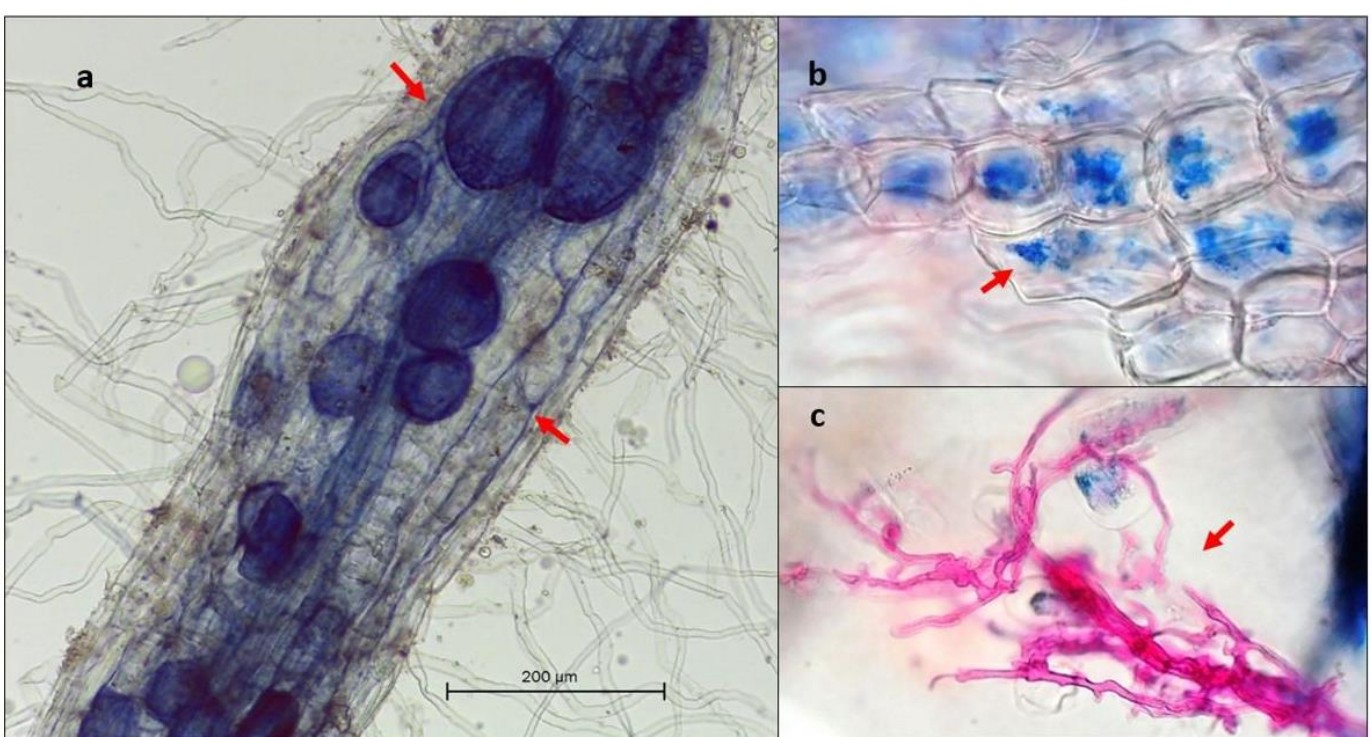

**Figure 1.** Arbuscular mycorrhizal structures present in cocoa roots in San Martin, Peru. (**a**) Vesicles and hyphae, (**b**) arbuscules, and (**c**) extraradical hyphae (red arrow).

### 3.4. About Easily Extractable GSRP and Total GSRP

In relation to GRSP-EE, no differences were recorded between the plantations studied. However, GRSP-T levels in the soils of the 9- and 10-year-old plantations were significantly higher than those in the soils of the 3-year-old plantation (Table 2). Notably, in the soils of these two plots (9- and 10-year-old), the organic matter content was higher than that in the soil of the three-year-old plot. Some authors, such as Emran et al. (2012) and Kumar et al. (2018) [54,55], observed that soil organic matter is related to the glomalin content detected in the soil. It is possible that the organic matter content is positively correlated with the glomalin content because it is a protein with a high presence of carbon in its structure.

### 3.5. Phylogenetic Analysis of the Glomeromycota Fungi Found in the Cocoa Plantations Studied

In this study, 34 samples were molecularly identified using the 18S rDNA gene. The percentage of similarity with the species described in the NCBI database was reported and the accession numbers was assigned (Table A1). The phylogenetic tree (Figure 2) showed that the studied sequences were clustered in 4 groups distributed in the families Archaeosporaceae, Acaulosporaceae, Glomeraceae, and Paraglomeraceae. Fourteen sequences shared the family Archaeosporaceae as the closest common ancestor and were clustered in a clade at group 1 (green) of the phylogenetic tree. On the other hand, it can be seen that 6 sequences were clustered in group 2 (red) in the clade of *Acaulospora*, 3 sequences were clustered in group 3 (blue) in the clade of *Claroideoglomus,* and 10 sequences were clustered in group 4 (yellow) in the clade of *Paraglomus*. The sequence LMTMb9 (MW534448) was grouped separately and close to the family Archaeosporaceae; however, we can observe that it was found within Glomeromycota and outside external groups made up of *Mortierella polycephala* and *Endogone pisiformes*.

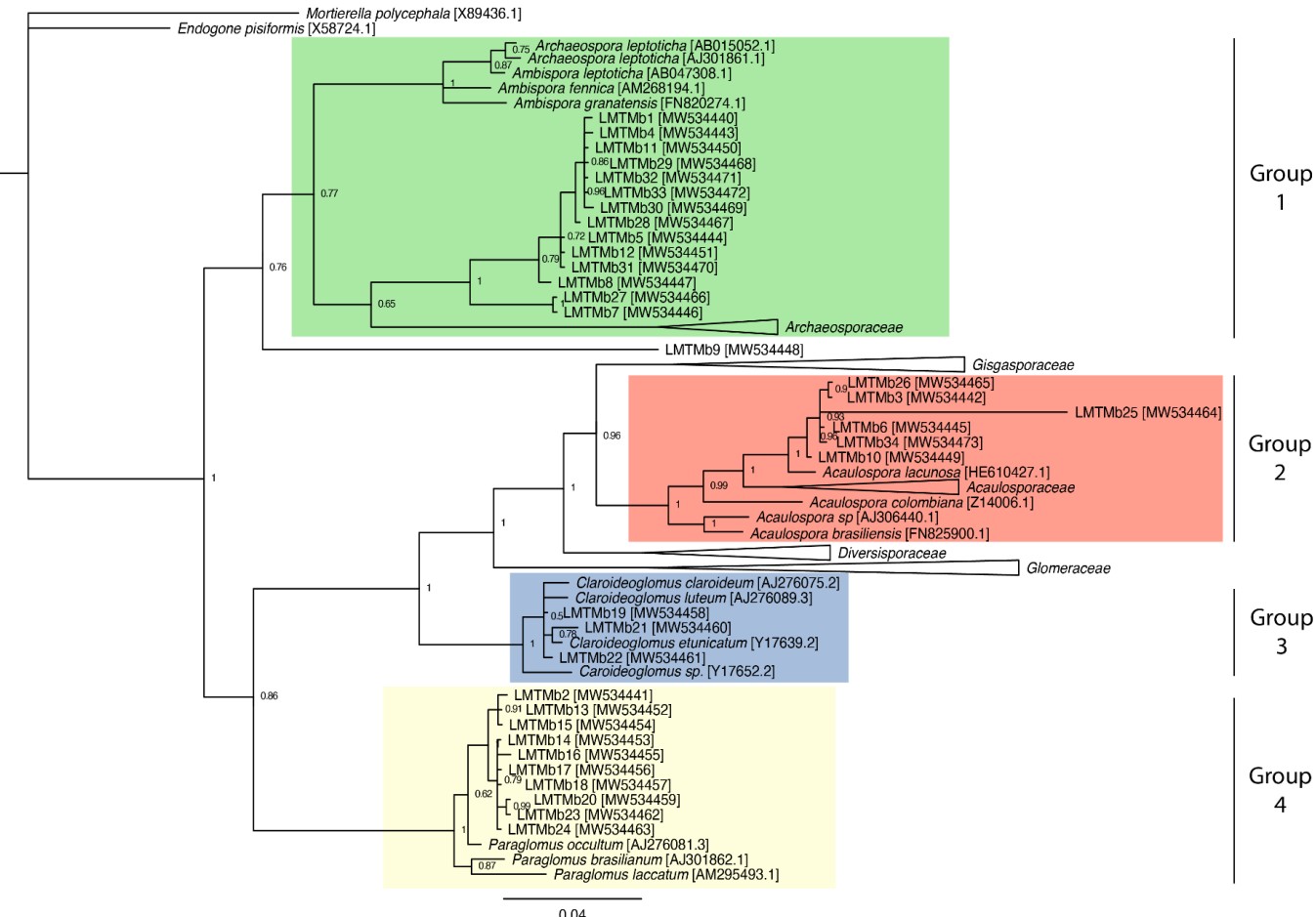

**Figure 2.** Phylogenetic tree based on SSU 18S sRNA sequences of Glomeromycota spores from three cocoa fields in San Martin, Tarapoto. The nodes of the phylogenetic tree correspond to posterior probabilities. Sequences were mainly clustered in 4 groups, except LMTMb9.

In Figure 2, the posterior probabilities of the different phylogenetic groups are translated into nodes, which correspond to the points where the bifurcations of the trees occurred and from which the groups or clades were defined. In our results, the nodes that defined clusters 2, 3, and 4 had a posterior probability of 1. The analysis indicated that there was 100% certainty that all sequences in that cluster formed a monophyletic cluster. In group 1, the posterior probability was lower, in this case we had 77% certainty of the monophyly of that group.

### 3.6. Glomeromycota Morphotypes in Cocoa Plantations

Figure 3 shows some of the morphotypes of the Glomeromycota fungi studied in the cocoa plantations, highlighting specimens of the families Glomeraceae, Acaulosporaceae and Paraglomeraceae, as well as auxiliary cells from the family Gigasporaceae.

### 3.7. Correlations between Biological Variables Measured

According to the Pearson correlations, the age of the plantation was negatively and highly correlated with the number of Glomeromycota spores ($p = -0.0079$ **) and very highly and significantly correlated with GRSP-T ($p = 0.0008$ ***). The number of coils and active hyphae was significantly correlated ($p = 0.0334$ *), as was total mycorrhizal colonization and GRSP-EE ($p = 0.0130$ *).

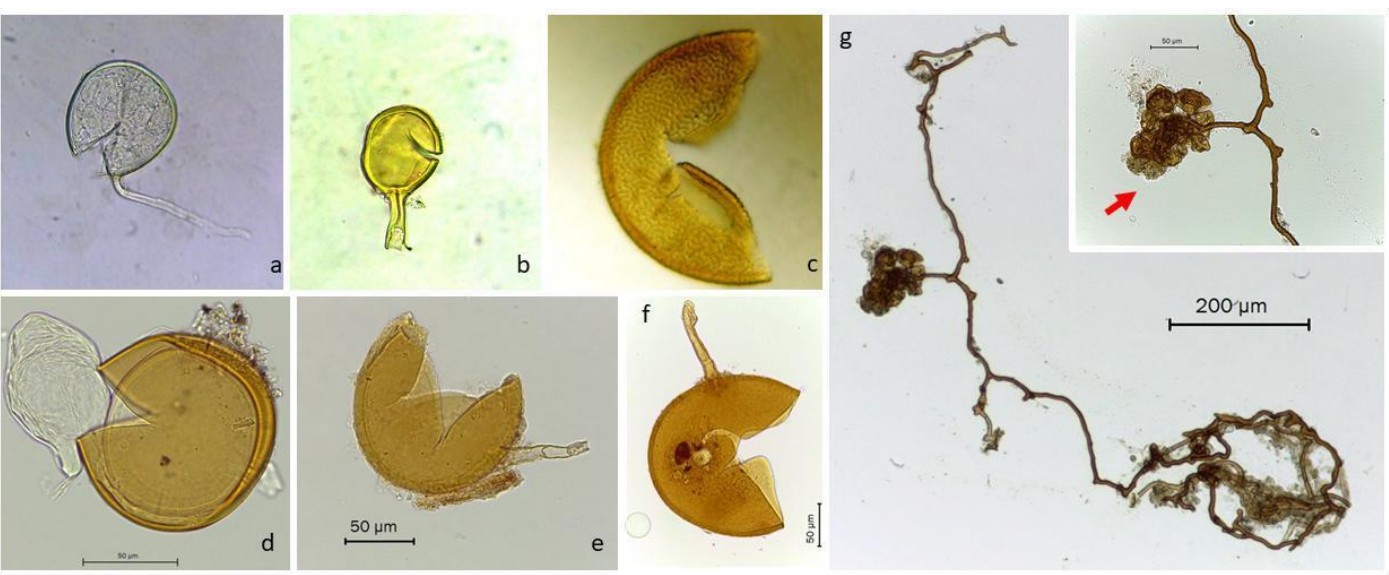

**Figure 3.** Some Glomeromycota morphotypes from the cocoa fields studied in San Martin, Tarapoto plantations. Three years: spores belonging to the families Paraglomeraceae and Glomeraceae (**a**,**b**). Nine years: spores belonging to the family Acaulosporaceae (**c**,**d**). Ten years: spores from the families Glomeraceae and Claroideoglomeraceae (**e**,**f**). Auxiliary cells of Gigasporaceae (red arrow) and mycelium are shown in (**g**).

## 4. Discussion

### 4.1. Soil Cd Content

Our results show a range of 0.37–1.09 mg Cd kg$^{-1}$ in soil. In Peruvian Amazonian plantations, Huamani-Yupanqui et al. (2012) [9] obtained 0.53 mg kg$^{-1}$; Zug et al. (2019) [56] obtained 0.18 mg kg$^{-1}$, and Oliva et al. (2020) [10] obtained higher values in the range of 1.02–3.54 mg kg$^{-1}$. In the Amazon area, the amount of Cd in the soil could be the most relevant driver of Cd concentration in cocoa. The previous statement is supported by Ramtahal et al. (2016) [3], who found 0.3–1.7 µg Cd g$^{-1}$ in several plantations in Trinidad, and Sandoval-Pineda et al. (2020) [24] reported 27.3 mg Cd kg$^{-1}$ in Colombian plantations. Peruvian legislation admits a maximum of 1.4 mg Cd kg$^{-1}$ in soil. Our results indicate that Tarapoto soils contain the amount of cadmium allowed for agricultural soils and would be suitable for growing cocoa.

### 4.2. Foliar and Seeds Cd Content

Foliar Cd values ranged between 1.58 and 2.00 mg kg$^{-1}$, close to the range reported by Gramlich et al. (2016) [23] in cocoa plantations in Bolivia (0.24–1.75 mg kg$^{-1}$) and that found by Zug et al. (2019) [56] (0.72 (mg kg$^{-1}$) in the Peruvian Amazon. However, our values were higher than those found by Huamani-Yupanqui et al. (2012) [9] in similar cocoa plantations (0.21 ppm) and those reported by Arévalo-Gardini et al. (2017) [1] in Cuzco (0.23 µg g$^{-1}$). Oliva et al. (2020) [10] found 0.49 and 2.53 mg kg$^{-1}$ in different parts of the plant, including the leaves. Conversely, our results were below those reported by Ramtahal et al. (2016) [3]. They obtained values of up to 5.21 µg g$^{-1}$ in cocoa leaves in plantations in Trinidad, quantities considered inappropriate for cocoa cultivation. Excessive application of fertilizers with HMs, high Cd content in parental soil material, or occasional floods that could have dragged polluting debris may have influenced the Cd plant content registered in this study.

We detected 0.3–0.63 mg Cd kg$^{-1}$ in cocoa beans, close to the range detected by Gramlich et al. (2016) [23] (0.14–0.29 mg kg$^{-1}$) and over [1] the range reported by Cuzco (0.17 µg g$^{-1}$). The same author found higher Cd seed concentrations in Tumbes plantations (1.78 µg g$^{-1}$) than Zug et al. (2019) [56] in Huanuco (2.46 mg kg$^{-1}$). Our results are below

the limit (1.5 mg kg$^{-1}$) allowed for final products of cocoa, which suggest that the use of Tarapoto cocoa seeds could provide final products without risk for nutrition and food.

The variety of cocoa studied in this work, clone CNN-51, and some of its hybrids have the lowest capacity to bioaccumulate Cd [1]. According to Meter et al. (2019) [57], the permissible content of Cd in cocoa beans for commercialization should be within the range of 0.5 to 1.1 mg kg$^{-1}$. Our results indicate that although the cocoa beans had the highest Cd value recorded (0.63 mg kg$^{-1}$) in the 10-year-old plantation, it was still within the range of values allowed for marketing cocoa products, and it may not represent a health risk. The use of the CCN-51 clone and its ability to bioaccumulate a smaller amount of Cd, in addition to the permissible levels of Cd in the soil, allow us to recommend its use for cocoa production, at least in Tarapoto.

### 4.3. Arbuscular Mycorrhizal Fungi Root Colonization

Succinate dehydrogenase (SDS), a tricarboxylic acid cycle enzyme reported to be an indicator of metabolically active fungal tissue in AMF [58], can be directly examined in tissues. Using this technique, AMF activity can be observed, unlike when trypan blue non-vital staining is employed in root tissue. Cuenca and Meneses (1996) [59] registered AMF colonization between 14% and 69% in cocoa plantations under conventional management. Iglesias et al. (2011) [27] recorded 34.8% colonization, 3.7 coils, and 5.9 vesicles in AMF-inoculated cocoa plants. Kähkölä et al. (2012) [22] recorded up to 36% mycorrhizal colonization and 23.5 arbuscules and hyphae in cocoa trials inoculated with Glomeromycota fungi, with a similar number of arbuscules as detected in this study. Gramlich et al. (2016) [23] registered colonization between 13% and 39% in conventional monoculture and agroforestry organic cocoa plantations in Bolivia. Pérez et al. (2019) [60] recorded mycorrhizal colonization of up to 60% in trials with Cd-tolerant cocoa plants, similar to this work.

Although none of the aforementioned works quantified mycorrhizal colonization by vital staining, our results show a significant number of arbuscules, coils, and active hyphae (Figure 1a,b), indicating that the cocoa plants sampled had physiologically functional symbiosis, more favored in the case of the 9-year-old plantation. The number of arbuscules detected and SDS activity are indicative of active and functional symbiosis, highlighting the fundamental role of arbuscules in the exchange of P and carbohydrates between the fungus and plant. In Table 2, it can be observed that active mycorrhizal colonization reached 63% while total colonization was up to 86%, indicating that most of the mycorrhizal structures were active in the three sampled plantations. Likewise, the mycorrhizal colonization values registered in this work were similar and slightly higher than those registered by Cuenca and Meneses (1996) and Pérez et al. (2019) [59,60] in tropical cocoa plantations, thereby confirming the microtrophy of cocoa plants.

### 4.4. Glomalin Related Soil Proteins

Glomalin is a glycoprotein produced by the hyphae of AMF [61]. Although the structure of glomalin has not been completely established [62], it is suggested that glomalin consists of monomeric structures linked through hydrophobic interactions [63]. However, in soil studies, glomalin is operationally defined by extraction and detection methods and known as glomalin-related soil protein [43,64], of which two fractions are known: easily extractable proteins (GRSP-EE) and total proteins (GRSP-T), as presented in this work.

The metal-binding capacity of GRSP varies with soil type and some physicochemical parameters, such as pH and redox potential [63,64]. Its potential to biostabilize heavy metals has been measured by several authors, such as Vodnik et al. (2008) [65]. They indicated that GRSP could sequester toxic elements through functional groups in its structure. GRSP has been detected in almost all soils in concentrations ranging from 2–15 mg g$^{-1}$ to more than 60 mg g$^{-1}$ [42,63]. Cornejo et al. (2008) [66] conducted work in metal-polluted soils (Cu 62–831 mg kg$^{-1}$) and found that the concentration of GRSP varied widely (6.6–37 mg g$^{-1}$), suggesting that the quantity of protein produced depends on the level of soil contamination. This amount of GRSP bound 1.44–27.5% of the total Cu content in the soil. A similar trend

was observed for Zn, highlighting the HM-remediating capacity of these proteins and AMF. Gonzalez-Chávez et al. (2009) [67] found that glomalin could form a complex with 0.08 mg Cd in the soil and sequester other HMs, such as Cu and Pb.

Singh et al. (2022) [68], working with several AMF isolates, found that glomalin removed Cu and Cd from an aqueous solution, with a mean glomalin-sequestering rate of 70.4% for Cu and 50.7% for Cd. Our results (Table 2) showed protein values between 8.30 and 13.75 mg g$^{-1}$, within the range described by Landa-Acuña et al. (2020), Wright and Upadhyaya (1998), and Cornejo et al. (2008) [19,20,66] for tropical forests. As Cornejo et al. (2008) [66] suggested, the amounts of GSRP recorded in this work could be related to the low quantities of Cd detected in the soil, considered not risky to health or fruit commercialization. Vodnik et al. (2008) [65] observed that GRSP could bind HM and was positively correlated with the total Pb soil concentration, suggesting that GRSP binds Pb. Hristozkova et al. (2017) [14] found that *P. peruviana* inoculated with AMF produced a higher amount of dry matter and had lower amounts of HMs (Cd and Pb) in the fruit compared to non-mycorrhizal plants.

In addition, *P. peruviana* produced 4.1 and 6.6 mg/g soil of GRSP-EE and GRSP-T, respectively, which suggested that the GRSP produced in the rhizosphere could have facilitated the lower incorporation of both HMs in the fruits of the mycorrhized plant. Ramtahal et al. (2012) [69] found that mycorrhized cocoas had lower amounts of Cd in the stem and leaves compared to non-mycorrhized plants. Although Ramtahal et al. (2012) [69] did not measure GRSP, their results suggest the possible phytoremedial role of AMF in cocoa plants with respect to Cd, as could be happening in the plantations of Tarapoto. The detection of both types of GSRP in the cocoa rhizosphere in this study indicates the capacity of this plant, as it is associated with mycorrhizae, to phytostabilize cadmium and heavy metals, a role described as mycorrhizoremediation [70].

### 4.5. Phylogenetic Analysis of Glomeromycota Fungi Found in the Cocoa Plantations Studied

In the phylogenetic tree shown in Figure 3, we determined the presence of four groups using sequences obtained from our samples. In group 1, the sequences were closely related to Archaeosporaceae, a family within the order Glomerales described in 2001 based on ribosomal DNA sequences [71]. This family is recurrent in soils and crops in South America, as reported by Schüßler and Walker (2019) [72], who discovered the new species *Archaeospora ecuadoriana*. This family has also been found in the neotropic [73] area in which this study was conducted. A BLAST search showed high similarity (>98%) between the sequences in group 1 to those previously reported from a global collection of AMF that associates with ancient liverwort [74] and hornwort species [75]. Interestingly, group 1 was also similar (>98%) to AMF collected from a tropical area in southern Ecuador [76].

In group 2 from Figure 3, our sequences clustered with members of the family Acaulosporaceae. According to Stürmer and Kemmelmeier (2021) [73], species from this family have been reported as the most abundant in the neotropics, who found 47 species of Acaulospora that represented 79% of species within this family, thus making it one of the most prevalent groups in this region. A very close example is *Acaulospora flavopapillosa*, a new fungus in the class Glomeromycetes from a coffee plantation in Peru, described using morphological observation tools in addition to analysis based on ribosomal DNA sequences [77]. A BLAST search showed the sequences in Group 2 were similar (>98%) to AMF samples collected from different regions in Ecuador [76].

In group 3 from Figure 3, our samples were closely related to species of the genus *Claroideoglomus*, whose presence has been evidenced in cocoa soils with high concentration of cadmium. The author of a previous study suggested that this genus is a potential candidate for the development of strategies to mitigate soils with Cd [24]. As in group 4 from Figure 3, our sequences were closely related to species of the genus *Paraglomus*. In the region of San Martín State in Peru, the presence of *Paraglomus occidentale*, a new species of AMF, has been described [78]. Furthermore, via BLAST search, sequences of both groups 3 and 4 from Figure 3 showed high similarity (>99%) with AMF described in

a study assessing AMF diversity in the restoration of a Riparian Forest in California [79]. Additionally, similar sequences (>98%) to group 4 have been found in Venezuela [80] and in forests in Ecuador [81]. Our sequence LMTMb9 (MW534448) was grouped separately from known families and may represent a new taxon. It showed similarity via BLAST (>98%) to samples collected from the rhizosphere of *Coccoloba uvifera* in Venezuela [80].

Molecular identification of AMF is challenging. In addition, many aspects of their basic biology remain unknown owing to their small size, difficulties in cultivation without a host plant (obligate biotrophy), and high genetic polymorphism [82,83]. Methodologies for the molecular identification of mycorrhizal fungi have been used in a similar way as in our study, as is the case of Rimington et al. (2018) [74], who used the TOPO TA cloning kit (Invitrogen) for the identification of mycorrhizal fungi in ancient liverwort. This author considers that it is necessary to develop a second PCR-type nested PC or semi-nested PCR in order to more accurately identify Glomeromycota fungi. Likewise, we consider that it is necessary to explore new methods for the identification of AMF, such as, for example, spore proteomic biotyping using MALDI-TOF-MS [84].

*4.6. Glomeromycota Morphotypes in Cocoa Plantations*

Regarding the number of spores observed, our results were close to those of Vestberg and Assefa (2015) [85], who found that the density of spores varied between 280 to 610 spores 100 $g^{-1}$ dry soil in agricultural systems. In conservative and semiconservative cocoa plantations, Pacheco et al. (2022) [86] registered of 300–400 spores 100 $g^{-1}$ dry soil. In this study, we registered 13.4- 131.8 spores 100 $g^{-1}$ dry soil. The slight difference between our results and those cited above could be due to cultivation practices and the edaphoclimatic conditions of the plantations, in addition to particular characteristics of the physiology and affinity of AMF to the cocoa variety of this study.

Morphological analyses indicated the presence of the families Paraglomeraceae (Figure 3a), Glomeraceae (Figure 3b,e), Claroideoglomeraceae (Figure 3f), and Acaulosporaceae (Figure 3c,d) in Tarapoto plantations. Our results coincide with those described by Cuenca and Meneses (1996) [59], who observed the predominance of the families Acauloporaceae and Glomeraceae in Venezuelan cocoa plantations, and the family Gigasporaceae to a lesser extent. We observed auxiliary cells (Figure 3g), which are diagnostic structures of the Gigasporaceae family, and although some spores were identified morphologically, the proportion of specimens of this family was much lower than that of the previously named families. Kramadibrata (2009) [87] also found similar genera in plantations in Indonesia and Ecuador. Likewise, Edy et al. (2019) and Nurhalisyah et al. (2020) [88,89] observed the predominance of the families Glomeraceae, Gigasporaceae, and Acaulosporaceae in Indonesian cocoa plantations and Rincón et al. (2020) [90] in cocoa plantations in the Ivory Coast.

Sandoval-Pineda et al. (2020) [24], studying cocoa plantations of Colombia, found that the diversity of Glomeromycota fungi was diminished by high concentrations of Cd in the soil; however, they observed the dominance of Diversisporaceae (*Diversispora spurca*), Glomeraceae (*Rhizoglomus* spp.), and Claroideoglomerceae (*Claroideoglomus etunicatum*) in these soils. These authors suggested that the morphospecies detected were potential candidates for the development of mitigation strategies in soils with Cd and/or for the elaboration of inoculants based on Glomeromycota fungi. Some of the morphotypes identified by morphology coincided with the Glomeromycota families detected by molecular analysis. However, the current work can be considered as a preliminary study since we continue carrying out morphological descriptions that will be complemented by subsequent molecular analyses.

*4.7. Correlations between Biological Variables*

The agronomic management of fields may affect the abundance of spores since according to our results and those of Tchabi et al. (2008) [91], the number of spores decreased with the longer cultivation time. Other studies described that a longer cultivation time favored the production of total glomalin, an effect that has been described by

Bedini et al., (2009) [92], who observed that conservation management favored soil aggregation, the total length of hyphae, and the volume of mycorrhized root of the crops, which may be related to the production of GRSP. It is known that mycorrhizal colonization is made up of internal structures such as vesicles, arbuscules, and coils, together with the fungal mycelium that extends outside the root [93]. An active mycorrhizal colonization should have abundant structures typical of the fungus when colonizing the root, as was the case in the current study. The correlation between the number of coils and active hyphae, together with that of the total mycorrhiza and GRSP-EE, suggests that there is an adequate mycorrhizal colonization process that favors the production of GRSP-T and GRSP-EE. This relationship was corroborated by Liang et al. (2021) [94], who described a positive relationship between GRSP production and mycorrhizal colonization, together with the abundance of internal structures, as in our study.

## 5. Conclusions

Our research demonstrated that mycorrhizal fungi belonging to the families Archaeosporaceae, Acaulosporaceae, Glomeraceae, and Paraglomeraceae in the plantations studied could be used for the preparation of inoculants applied in the nursery. Mycorrhized cocoa plants were able to produce GRSP in quantities similar to those of other studies where the HM (as cadmium) sequestration capacity was tested. We suggest continuing research in this area in order to make use of AMF in improving the quality of cocoa cultivation and its products.

**Author Contributions:** Conceptualization, M.T., D.Z.-D. and K.O.-G.; Methodology, M.T., E.O.-O. and B.L.-A.; Software, E.O.-O.; Formal analysis, M.T., B.L.-A., D.Z.-D. and E.O.-O.; Investigation, B.L.-A., R.C. and A.G.-P.; Resources, D.Z.-D.; Writing—original draft preparation, M.T. and B.L.-A.; Writing—review and editing, M.T., D.Z.-D. and B.L.-A.; Project administration, D.Z.-D.; Funding acquisition, D.Z.-D. All authors have read and agreed to the published version of the manuscript.

**Funding:** Bernabé Luis-Alaya was funded by scholarship grant No 177-2015-FONDECYT from the National Council of Science, Technology and Technological Innovation of Peru. Project No 009-2017-FONDECYT from the National Council of Science, Technology and Technological Innovation of Peru.

**Institutional Review Board Statement:** Not applicable.

**Informed Consent Statement:** Not applicable.

**Data Availability Statement:** The 34 sequences from this study are available in the National Center for Biotechnology Information (NCBI) under accession numbers MW534440 to MW534473.

**Acknowledgments:** The authors thank Centro de Innovación del Cacao, Peru (CIC) and Cooperativa Agraria Cafetalera Oro Verde for technical support. We also thank Auxi Toro Garcia, translator, for revision of the English language in this manuscript. Our sincere thanks are extended to Daniela Landa, Andi Solorzano, Kenyo Ramos, Pilar Echaiz, Andrea Delgado, Jorge Mendoza, David Saravia-Navarro, Yuri Arévalo, Jorge Rios, and Claudia Valencia for administrative and technical support.

**Conflicts of Interest:** The authors declare no conflict of interest.

## Appendix A

**Table A1.** Similarity (percentage), accession number, and closest molecular identification using 18S rDNA gene of Glomeromycota spores from three cocoa fields in San Martin, Tarapoto via BLAST against the NCBI database.

| Cocoa Field | Lab Sequence | Closest Molecular Identification Using 18S rDNA Gene | Accession Number | Similarity (%) |
|---|---|---|---|---|
| 10 years | LMTMb1 | *Glomeromycotina* spp. | MW534440 | 98.75% |
| 10 years | LMTMb2 | Uncultured *Glomus* | MW534441 | 99.62% |
| 10 years | LMTMb3 | Uncultured *Acaulospora* | MW534442 | 99.62% |
| 10 years | LMTMb4 | *Glomeromycotina* spp. | MW534443 | 98.62% |
| 10 years | LMTMb5 | *Glomeromycotina* spp. | MW534444 | 98.62% |

**Table A1.** *Cont.*

| Cocoa Field | Lab Sequence | Closest Molecular Identification Using 18S rDNA Gene | Accession Number | Similarity (%) |
|---|---|---|---|---|
| 10 years | LMTMb6 | Uncultured *Acaulospora* | MW534445 | 99.75% |
| 10 years | LMTMb7 | *Glomeromycotina* spp. | MW534446 | 97.60% |
| 10 years | LMTMb8 | *Glomeromycotina* spp. | MW534447 | 98.24% |
| 10 years | LMTMb9 | *Glomeromycotina* spp. | MW534448 | 98.12% |
| 10 years | LMTMb10 | Uncultured Glomeromycotina | MW534449 | 99.62% |
| 10 years | LMTMb11 | *Glomeromycotina* spp. | MW534450 | 98.74% |
| 10 years | LMTMb12 | *Glomeromycotina* spp. | MW534451 | 98.62% |
| 3 years | LMTMb13 | Uncultured *Glomus* | MW534452 | 99.75% |
| 3 years | LMTMb14 | Uncultured *Glomus* | MW534453 | 100.00% |
| 3 years | LMTMb15 | Uncultured *Glomus* | MW534454 | 99.75% |
| 3 years | LMTMb16 | Uncultured *Glomus* | MW534455 | 100.00% |
| 3 years | LMTMb17 | Uncultured *Glomus* | MW534456 | 100.00% |
| 3 years | LMTMb18 | Uncultured *Glomus* | MW534457 | 100.00% |
| 3 years | LMTMb19 | Uncultured *Claroideoglomus* | MW534458 | 100.00% |
| 3 years | LMTMb20 | Uncultured *Glomus* | MW534459 | 99.87% |
| 3 years | LMTMb21 | Uncultured *Claroideoglomus* | MW534460 | 100.00% |
| 3 years | LMTMb22 | Uncultured *Glomus* | MW534461 | 100.00% |
| 3 years | LMTMb23 | Uncultured *Glomus* | MW534462 | 99.87% |
| 3 years | LMTMb24 | Uncultured *Glomus* | MW534463 | 100.00% |
| 10 years | LMTMb25 | Uncultured *Glomus* | MW534464 | 100.00% |
| 10 years | LMTMb26 | Uncultured *Acaulospora* | MW534465 | 99.62% |
| 10 years | LMTMb27 | *Glomeromycotina* spp. | MW534466 | 97.60% |
| 10 years | LMTMb28 | *Glomeromycotina* spp. | MW534467 | 98.87% |
| 10 years | LMTMb29 | *Glomeromycotina* spp. | MW534468 | 98.74% |
| 10 years | LMTMb30 | *Glomeromycotina* spp. | MW534469 | 98.62% |
| 10 years | LMTMb31 | *Glomeromycotina* spp. | MW534470 | 98.73% |
| 10 years | LMTMb32 | *Glomeromycotina* spp. | MW534471 | 98.85% |
| 10 years | LMTMb33 | *Glomeromycotina* spp. | MW534472 | 98.85% |
| 10 years | LMTMb34 | Uncultured *Acaulospora* | MW534473 | 99.62% |

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
