# Peer review of "Evaluation of the Presence of Arbuscular Mycorrhizae and Cadmium Content in the Plants and Soils of Cocoa Plantations in San Martin, Peru"

_diversity, doi:10.3390/d15020246_

Round 1
Reviewer 1 Report
The manuscript presented by Luis-Alaya et al. consists in a report about the presence of arbuscular mycorrhizal fungi associated with cocoa plants of different ages of plantation, which tries to associate their potential for Cd remediation in Peruvian soils. The study presents several problems that I consider important to take into consideration, being my present evaluation unfavorable regarding its possible publication. Please, see my point-by-point arguments.
-In general, the study is carried out on variables that according to the experimental description it is impossible to imagine the design of the sampling. Sometimes five units were mentioned, in others three, some compound, others not. This lack of uniformity avoids for the search of relationships between the response variables (section 2.2 and 2.3).
- The wording must be greatly improved, especially formal aspects such as abbreviations, which are sometimes made, not used, or defined in different places in the text. Take “heavy metals”, for example, which should have been defined in L44. There are, however, whole sections that are difficult to understand.
-It is recognized that neither the soil nor the plant materials analyzed present Cd problems, so the focus of the study, oriented to the role of AMF (see title) in reducing Cd bioaccumulation, is not logical. Moreover, considering the acidity levels of the soils, it is logical to assume a high mobility of the metal (not determined or not explained), which should rather be analyzed according to the role of the AM symbiosis decreasing the translocation of the contaminant (translocation indices for instance).
-It is equally acceptable that the relationship with the Cd mobilization to the shoots can be mediated by the composition of the AMF community, but this aspect is only descriptive, and rather at an elementary level, which does not show the expression of diversity indices or structure of the AMF communities. This would have been desirable at least in the section referring to the description of the presence of various morphotypes of AMF.
-The discussion turns out to be rather descriptive, and comparative with other studies not entirely related. In this sense, the study must include a most deep degree of analysis considering the scalar response values, including a multivariate perspective or through their correlations, for instance.
-Finally, it is concluded that the important role of glomalin may be the immobilization of Cd in the soil, given the values found, but this should be analyzed according to the content of Cd that is presented in the fraction obtained from GRSP.
Other comments:
1.-The abstract is extremely large, with a pronounced section of background not necessarily important to highlight at this level.
2.-L52, no need to detail the same unit twice.
3.-L57, change 0,5 by 0.5
4.-L74-75, Must be oriented only to AMF.
5.-L79, exerts?
6.-L82, here and elsewhere is confused when referring to the “fungus” (AMF) with respect to when referring to “symbiosis” (AM). For example, see L55-57.
7.-L93, was previously defined as AMF, or as AM fungi? Please unify to a single format.
8.-L118-119. The variety of cocoa is mentioned in at least three different ways in the document (CCN51, CCN 51, CCN-51).
9.-Section 2.6. The methodologies described for the different GRSP fractions, especially EE, are different from those originally described and in other studies, according to autoclaving time, pH of the extractant solution and concentrations. Perhaps, this level of detail was not necessary since the methodology is widely described in many articles.
10.-L216, why the parentheses?
11.-In section 2.7 finally the protocol is mentioned only for spores, but not for hyphae (as mentioned in the same text..., L215).
12.-L279, with these pHs it is necessary to know the availability of the metals, presumably high. It should be more specific in what HM fraction was the specifically reported.
13.-Table 1. The reported values seem to have no replications, but they are assumed as experimental data, not just descriptive, which does not seem correct.
14.-L309-322. The presentation of results seems to be mixed with discussion of the data.
15.-Table 3 present a very limited volume of data that do not justify its presentation by itself. Being a component more associated with AMF, it should be part of Table 2.
Reviewer 2 Report
This is my feedback to the article “Potential of arbuscular mycorrhizae for cadmium remediation in organic cocoa plantations in Peru”.
The authors have planned their investigation according to the problem they selected. The article's title accurately describes what it discusses. The research results are relevant and crucial for using arbuscular mycorrhizae to reduce heavy metal contamination in the agricultural sector and the work gives intriguing material in this regard. However, I suggest some revisions to improve the manuscript’s quality.
-English language must be revised by an expert.
-Line 35: In the keywords section you should highlight the important key elements presented in the abstract, however, I don’t understand the use of the 18s gene that was not mentioned in the abstract.
In the same line; please correct the 18s gene.
Use only one name of the cocoa plant (the common name or the scientific one).
-Line 50: Cite the author in “as indicated by” and then the reference according to MDPI style.
-Line 56: Please correct Latinamerian.
-Line 83: Please cite the author’s name and then the reference number. The same is in lines 92, 145, 166, 181, 192, 200, 203, 206…. And so on. Please revise them throughout the manuscript.
-From lines 145 to 147: Reformulate the paragraph and cite the author’s name as mentioned above.
-Adjust tables form and structure according to the MDPI style. The same with the different figures.
-Lines 205-206: substitute mg protein / g of soil with mg protein per gram of soil. In the same line, please provide the equation of the calibration curve.
-Line 249: Phylogenetic tree is repeated twice. Please correct the typo.
-Line 302-304: Adjust the space between the paragraphs.
-In the discussion section: “Our results show a range of 0.37-1.09 mg Cd.kg−1 in soil. In Peruvian amazonian plantations [9] obtained 0.53 mg.kg−1 ; [55], 0.18 mg. kg−1, and [10] obtained higher values in a range of 1.02-3.54 mg. kg−1” . The presented references are misplaced which generated confusion in understanding the content. Please put the references in the corresponding places and according to MDPI style. Moreover, the discussion sections should be linked to each other proving an interpretation of your results with previous studies.
-Lines 590-593: I don’t find a necessity to put table A1 in the manuscript as an appendix. In my suggestion, this table should be removed from the manuscript or moved to supplementary materials if provided.
In summary, this field of research is important and I suggest accepting the article but only after revisions on the English language and the form of the manuscript, and the references according to MDPI style to fit the Journal’s recently published articles.

Reviewer 3 Report
The manuscript was studied the Cd contents, soil properties, and AMF colonization and communities in the three Tarapoto plantations. The authors concluded that the potential of arbuscular mycorrhizae for cadmium remediation in organic cocoa plantations in Peru. However, I did not find the relationship between Cd remediation and AM symbiosis and the potential of AMF for cadmium remediation. Thus, I think the title and topic should be revised according the text.
The authors showed that “Peru is the second largest producer of organic cocoa worldwide, with 1.7% of the world production of cocoa seeds”. But I did not find the data in the reference 1.
Line 66, the beneficial microorganism is arbuscular mycorrhizal fungi, not AM. Please distinguish.
Line 128, I think the soil sample is not rhizospheric soil according to the description.
Line 294, I did not understand the means of “* Low ** Medium *** High”.
Please check the format of the tables in the manuscript.
The AMF were just identified to families. Can you identified to species?
The Discussion should be improved. The data and context cannot be duplicate with the Results.
Please check the language throughout the manuscript.
Round 2
Reviewer 3 Report
The manuscript was revised according to the reviewers' comments and suggestions. However, some points should be understood, for example, please check the concept of rhiospheric soil.
